# The Quality of Fair Revaluation of Fixed Assets and Additional Calculations Aimed at Facilitating Prospective Investors' Decisions

**Sarfraz Hussain** [1,*], **Mohammad Enamul Hoque** [2], **Perengki Susanto** [3,*], **Waqas Ahmad Watto** [4], **Samina Haque** [2] **and Pradeep Mishra** [5]

1 Putra Business School, Universiti Putra Malaysia, Serdang 43400, Selangor, Malaysia
2 BRAC Business School, BRAC University, Dhaka 1212, Bangladesh
3 Department of Management, Universitas Negeri Padang, Padang 25131, Indonesia
4 Commerce Department, Bahauddin Zakariya University, Multan 60000, Pakistan
5 College of Agriculture, Jawaharlal Nehru Krishi Vishwavidyalaya, Powarkheda, Narmadapuram 461110, India
* Correspondence: pbs21204242@grad.putrabs.edu.my (S.H.); perengki@fe.unp.ac.id (P.S.)

**Abstract:** The main objective of this study is to find out why sugar companies' revaluation of their fixed assets has no direct financial impact. The purpose of this financial statement analysis of the sugar sector is to help potential investors make better decisions. It can also be used to address information asymmetries and alert investors. Fixed assets form a major part of a company's value. During 2013–2018, 19 selected enterprises of Pakistan's sugar sector adopted the International Accounting Standards Board's international accounting standard 16 for fixed assets. Ordinary least squares, fixed effects, and random effects methods were used as a static panel, a panel-corrected standard errors method was used for the robust standard error and the system generalized method of moments was used as a dynamic panel. The surplus had a negative impact on operative income on revaluation of fixed assets in sugar businesses. As expected, revaluation by fixed asset firms resulted in changes in potential outcomes, as measured by cash in operating income and revenue, both of which were extremely negative. The return on assets was also linked to revaluation balance. The debt over the proportion of assets resulted in a strong correlation between revaluations, which meant that motivation affected how the volatility in asset value reflected the revaluation. Relationships were generally worse and more uncertain for listed companies at a time of strong economic volatility. Investors should not consider such accounting justice. The price-earnings ratio had a beneficial effect on operative income. The statistics support the idea that external concerns help the revaluation of assets.

**Keywords:** revaluation; fixed assets; FASB; prospective investors; investment decisions; Financial Accounting Standards Board

## 1. Introduction

Financial statements are an important method of communicating corporate results to external investors, in addition to reflecting on a company's position [1]. Aaccording to the Financial Accounting Standards Board (FASB), the general purpose of financial reporting is to offer financial information about the reporting agency, which helps current and future owners, borrowers, and other lenders to make choices about whether to give resources to the organization. The purchase, sale, or retention of stock and loan equipment, as well as debt issue or settlement and other sources of debt, are examples of these choices [2]. However, investors' decisions to buy or sell securities depend on the expected return on equity. As financial statements represent the identity of a company, accounting treatment choices are based on the discretion of the management [3]. This collection is a platform

for enterprise financing content. The restoration of real assets has a major impact on the company's financial statements. According to the need for accounting knowledge, there are differences of opinion and financial differences between the management and the lenders [4]. Therefore, in this context, the accounting profession helps to close this gap and show the true situation [5].

Fixed assets contribute to operational revenue generation, which speaks of their crucial importance [6]. Therefore, there is a need for them at a reasonable price. The International Accounting Standards Board (IASB), which also discusses the need to integrate accounting standards, issued international accounting standard 16 (IAS16), the standard for fixed assets [7]. International Accounting Standard 16 (IAS 16) provides two models for assessing fixed assets: recording assets on a historical basis or revising them at market value [8]. When a corporation wants to revalue its properties, plants, machinery, and equipment, it is performed consistently so that the asset is not sold at a price that varies from the current valuation [9]. Basic other principles are also mentioned in the revaluation surplus (Summary of IAS 16). Information release relies on helping investors manage their investment portfolios. The problem proportional to information is resolved through a re-examination of capital assets, thus providing a means of transmitting information to lenders. It has been stated that the motivation behind the revaluation of Australian managers' assets is to show accurate and equitable financial returns. The analysis of real assets demonstrates the true value of fixed assets and thus tends to improve a company's stock valuation [10]. A variety of studies have demonstrated that the reassessment of plants and equipment is a significant means of increasing a company's potential success [11].

International Accounting Standard 16 (IAS 16) applies since January 2005 [12]. Pakistani businesses are often faced with the option to report fair value properties or revalue them at market value. Previously, a small investigation was carried out in the Pakistani context to see if there was an effect of the surplus on the revaluation of capital assets on the results of corporations' performance [13]. Hence, this analysis aimed to see the impact of a fixed asset revaluation on Pakistani companies' potential firm results. Only the sugar sector was selected. The sugar industry is a big business in Pakistan that flourished before partitioning. Production at this time has risen more than three times compared to that at independence time. The key assistance to this industry is the supply of limitless raw materials for manual invoicing. International Accounting Standard 41 (IAS 41), Agriculture, was the first standard directly covering the primary sector standard that is regulated and practiced in Pakistan. Therefore, the real issue is that fixed asset revaluation decides the potential output of companies in Pakistan's sugar market. The purpose of this analysis, therefore, was to clarify the need for financial statements, to understand how the appraisal affects current and prospective investors, and, more specifically, to describe the link between the revaluation of fixed assets and the company's possible success in Pakistan's sugar market. The analysis of the reassessment of fixed assets in Pakistan is worth considering because it can be an alternative to the one carried out in other countries. The findings would also assist managers and investors evaluate the value of asset revaluations and their effect on the successful appraisal of a [14].

## 2. Literature Review

The focus of the study of fixed asset revaluation is its impact on a firm's market value. Several studies are available that have evaluated the associated reassessment of capital assets and their effect on the results of a company's performance [14]. It was confirmed that the rising valuation of capital assets of UK companies had a positive correlation with potential yield improvements such as net income and cash from sales [15]. It was also observed that in the current year, the revaluation (recovery appraisal) was modest compared to the actual profit (prices). A study inspected New Zealand companies and found a good relationship with re-evaluation and return [16]. A previous study, conducted on the Swiss Stock Exchange, looked at the underlying asset market price dynamics for companies that were duplicated and found that foreign lenders and investors were concerned about these

companies' financial position, which strengthened their ideas [17]. A study examined Australian asset reassessment and share values and company value forecasts, focusing on nonmarket-based firms [18]. They considered revalued plants and machinery of greater pertinence. However, ref. [19] found in their earlier work on Spanish companies that asset revaluations were uncorrelated with the knowledge that was meaningful to investors [19].

Other authors assessed that the relationship between the assets was clarified by the same results, revaluations, and future operational results, and they acknowledged that market-based assessments had only partial evidence of this relationship [20]. Another point of view is that fixed asset revaluation has a negative effect on company profitability [21]. Some authors examined the correlation between the revaluation of capital assets and the potential success of companies [22]. Results showed that the revaluation of assets was performed purely to boost equity status and investors would not see it as practical. Therefore, they found a negative correlation between fixed asset revaluation and potential company profitability [23]. Revaluation is at the discretion of the administration, with researchers also trying to find an administrative purpose under asset revaluation [24]. Some authors studied Hong Kong administrators' impetus for an upward reassessment of capital assets [25]. The findings suggested that the managers' revaluation aimed to provide a fair measure of the market value of a company and to improve a company's borrowing ability [26].

Historical expenses versus market value of the two accounting methods calls attention to the appropriateness of each process [27]. As the approach to market valuation gives a true impression of the industry, the problem of intensified incentives for depreciation and management raises concerns regarding its suitability. A study tried to show that the historical cost accounting approach was highly suitable for the reassessment model [28]. When addressing the intersection point of the U.S. GAAP and IAS 16, analysts claimed that using market valuation was not a credible effort to demonstrate the real depreciation of properties, but instead compared market value and asset use [29]. However, as the choice of an accounting system relies on the discretion of the board, bias and controversies in this respect can be quite common [30]. Some findings also argued that if owners and shareholders assessed the real value of the properties by market value, other intangible elements were required for the accurate valuation of the securities [31]. It was reported that these businesses were reporting their investments at a fair market value with strong results on the market [32]. Upward revaluation indicates positive returns for businesses as creditors find a firm's business position good. By considering both hypotheses, the present research aimed to determine the impact of the reassessment of fixed assets on a prospective company's output in the sugar sector in Pakistan.

Pakistan's loan agreements are generally limited to total noncurrent assets and secured liabilities (based on the type of loan issued) [33,34]. The violation of the limit imposes costs mostly on a firm, as it is best forced to spend and finance it [35], including costly repayment or renewal costs of loans. It helps ease loan responsibilities as restoration increases the book value of aggregate tangible assets. A formal debt requirement to avoid those costs will thus increase the chances of reevaluation. This statement applies to the assumptions below.

**Hypothesis 1 (H1).** *Because of the effect of currency fluctuations (REER) on fixed assets value, firms frequently review fixed assets during high exchange rate hikes.*

**Hypothesis 2 (H2).** *Long-term investment has a positive relation with operational income.*

**Hypothesis 3 (H3).** *Companies update the market value of fixed assets more frequently and depict it as a negative association with operating income in their financial statements.*

**Hypothesis 4 (H4).** *The higher the debt-assets ratio, the more likely a firm is to re-evaluate its fixed assets.*

**Hypothesis 5 (H5).** *A more liquid position of a company affects its operational income.*

## 3. Research Methodology

*3.1. Data*

This research aimed to examine the impact of fixed asset revaluation on the future output of companies in Pakistan's sugar market. Pakistan Sugar Manufacturers Association has 22 members reporting to the State Bank of Pakistan. Only 19 firms were selected because of data processing issues and only 19 revalue their properties against market prices. The officially obtained quantitative data published by the State Bank of Pakistan were collected from the financial statements of the companies. The research time frame was 2013–2018.

*3.2. Variables*

The analysis aimed to assess the impact of the revaluation of fixed assets on the possible success of Pakistani sugar firms. The success of a company was calculated by its operating revenue. A direction was changed to arrive at the correct conclusion. Table 1 describes the quantities, objects, and methods of measuring them.

$$OI = \text{Operating Income (Gross Income} - \text{Operative Expenses)}$$

**Table 1.** Descriptive statistics.

|  | OI | EXR | LTI | REFA | DAR | CIR | PER | ATR | ROA |
|---|---|---|---|---|---|---|---|---|---|
| Mean | 9.271 | 4.655 | 4.676 | 13.450 | 0.584 | 5.749 | 17.844 | 0.801 | 4.607 |
| Median | 11.294 | 4.664 | 0.000 | 13.884 | 0.657 | 1.864 | 3.901 | 0.771 | 2.820 |
| Maximum | 13.577 | 4.706 | 14.477 | 16.021 | 1.248 | 56.817 | 490.791 | 2.241 | 18.084 |
| Minimum | 0.000 | 4.593 | 0.000 | 0.000 | 0.000 | 0.000 | 0.000 | 0.000 | 0.000 |
| Std. Dev. | 4.706 | 0.039 | 5.923 | 2.363 | 0.322 | 10.573 | 57.821 | 0.459 | 4.636 |
| Skewness | −1.328 | −0.372 | 0.528 | −4.850 | −0.589 | 3.131 | 6.070 | 0.559 | 1.128 |
| Kurtosis | 3.081 | 1.796 | 1.377 | 28.193 | 2.449 | 13.472 | 44.569 | 3.369 | 3.201 |
| Observations | 114 | 114 | 114 | 114 | 113 | 113 | 113 | 114 | 114 |

As we previously said, revenue refers to earnings before deducting any expenditures or expenses. Operational income, on the other hand, is a company's profit after deducting operating expenditures, which are the costs associated with doing regular operations. By removing interest and taxes, operational income aids investors separate the profits from the company's operating performance [36].

$$EXR = \text{Exchange Rate i.e., REER (Real Effective Exchange Rate)}$$

$$LTI = \text{Long Term Investment}$$

$$REFA = \text{Re-Evaluation of Fixed Assets}$$

$$DAR = \text{Debt-to-Assets Ratio (Short-Term and Long-Term Liabilities)/(Noncurrent Assets + Current Assets)}$$

$$CIR = \text{Cash-to-Income Ratio (Operating Cash Flow/Net Profit)}$$

$$PER = \text{Price-Income Ratio (Market Price per Share/Basic Income per Share)}$$

$$ATR = \text{Assets Turnover Ratio (Net Sales/Total Assets)}$$

$$ROA = \text{Return on Assets \{Net Profit/(Current Assets Plus Noncurrent Assets)\}}$$

$$EXR \times DAR = \text{REER (Real Effective Exchange Rate} \times \text{Debt-to-Assets Ratio)}$$

*3.3. Model*

It is clearly observed in Figure 1; that the connection between capital asset revaluation and prospective business performance was determined using the accompanying method. The operational profitability should influence future production changes. The study aimed

to see how revaluation factors affect business operational revenue using a panel data analysis of cross-sectional time-series data from 2013 to 2018. Operating income (OI) was used as a counter variable with a combination of variables; hence, operating income can be interpreted as follows:

$$OI = F(LongTermInvestment, \ ReevaluationFixedAssets, \ DebtsAssetsRatio, \ CashIncomeRatio, \ AssetsTurnoverRatio, \ RetrunOnAssetsRatio, \ PriceEarningRatio, \ ExR) \tag{1}$$

**Static Panel Model**

The simple linear regression equation is:

$$Y_{it} = \alpha_{it} + \beta_{it}(X) + \varepsilon_{it} \tag{2}$$

Static linear models stand accessible in the subsequent 2nd and 3rd empirical equations:

$$OI_{it} = \beta_{it} + \beta_1(LTI_{it}) + \beta_2(REFA_{it}) + \beta_3(DAR_{it}) + \beta_4(CIR_{it}) + \beta_5(PER_{it}) + \beta_6(ATR_{it}) \\ + \beta_7(ROA_{it}) + \beta_8(EXR_{it}) + \varepsilon_{it} \tag{3}$$

**The static model exchange rate with debt to assets ratio as interaction effect is:**

$$OI_{it} = \beta_{it} + \beta_1(LTI_{it}) + \beta_2(REFA_{it}) + \beta_3(DAR_{it}) + \beta_4(CIR_{it}) + \beta_5(PER_{it}) + \beta_6(ATR_{it}) + \beta_7(ROA_{it}) \\ + \beta_8(EXR_{it}) + \beta_9(EXR * DAR_{it}) + \varepsilon_{it} \tag{4}$$

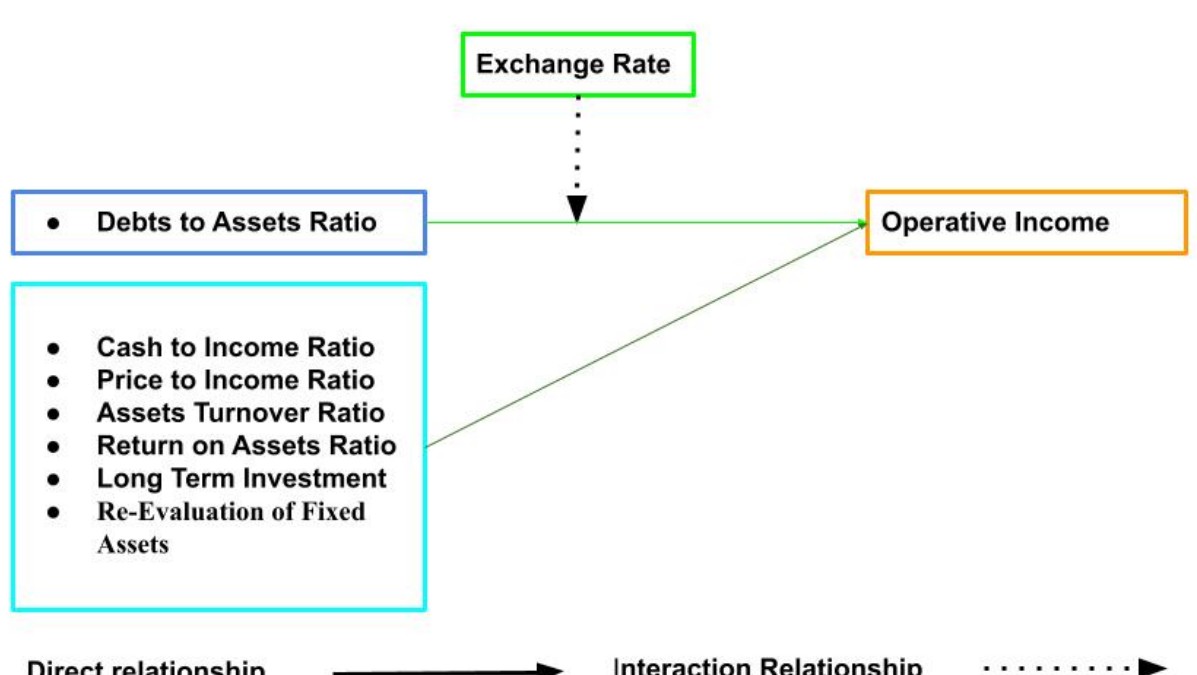

**Figure 1.** Conceptual framework.

Everywhere, $i$ ($i = 1 \ldots 19$) is the intercept of the firm, $t$ ($t = 2013$–$2018$) is the year studied, and they are coefficients for each regression variable, including the term error. Static panel models were evaluated using a variety of methods, including pooled ordinary minimum squares (PLS), random effects (RE), and fixed effects (FE), including N-firm specific intercepts. Fixed-impact models view the relationship between input and output variables in different ways, considering that Pakistani companies have unique features that affect the association of related variables. Random impact patterns, on the other hand, show random differences between organizations that are unrelated to input factors. According to the Brush and Pagan multiplication tests, the random impact model is suitable between ordinary least square and random effects, while the Hasman test defines the best model from other double models. Finally, to solve problems of self-correlation and heteroscedasticity, the 5th to 7th regression models were used, especially a regression with

a two-stage system generalized method of moments, which required a modification of the estimated model. The panel-corrected standard errors of the table's 3rd and 4th column panel series was used for the standard error.

Dynamic Panel Model

Many aspects of business, banking, economics, and finance are character-driven, and panel data setups are used to agree on changes. For the constant estimate of additional parameters, it is critical to enable dynamics in the main process. Carrying a lagged dependent variable with the regressors describes the dynamic linkages, i.e.,

$$Y_{it} = \delta Y_{i,t-1} + \beta x_{it} + \mu_{it} \tag{5}$$

$$OI_{it} = \beta_{it} + \beta_1(LTI_{it}) + \beta_2(REFA_{it}) + \beta_3(DAR_{it}) + \beta_4(CIR_{it}) + \beta_5(PER_{it})$$
$$+ \beta_6(ATR_{it}) + \beta_7(ROA_{it}) + \beta_8(EXR_{it}) + \mu_{it} \tag{6}$$

**The dynamic exchange rate with the debt-to-assets ratio as an interaction effect is:**

$$OI_{it} = \beta_{it} + \beta_1(LTI_{it}) + \beta_2(REFA_{it}) + \beta_3(DAR_{it}) + \beta_4(CIR_{it}) + \beta_5(PER_{it}) + \beta_6(ATR_{it}) + \beta_7(ROA_{it})$$
$$+ \beta_8(EXR_{it}) + \beta_9(EXR * DAR_{it}) + \mu_{it} \tag{7}$$

where $\delta$ is a scalar, $X_{it}$ is $1 \times K$, and $\beta$ is $K \times 1$ with K a real number. $\mu_{it}$ goes with the term's one-way disturbance component model, $\mu_{it} = \lambda_i + \varepsilon_{it}$. $\mu_{it}$ is merged into $\lambda_i$ and $\varepsilon_{it}$, where $\lambda_i$ has an individual specific effect to cover specific dissociation and $\varepsilon_{it}$ is the term error. The empirical model is considered to promote investment variables. Because equity can range from investment to firm equity or to debt in both directions and vice versa, these restrictions can be synchronized through the error term. Time-oriented firm individualities (unobserved specific effects, $\lambda_i$), such as demographics and geography, can remain integrated through descriptive variables. The presence of the lagged measured variable leads to autocorrelation. There are at least two reasons for measuring a short period ($t = 6$), then measuring a firm in a panel dataset ($n = 19$): simultaneous error control makes it possible to predict some variables (associated). The firm's specific dummy variables cannot be used while controlling the firm's exact effects, which is due to the dynamic gathering of regression calculations. According to the theory, the experimental model promotes investment factors. It is possible to adjust these limits by using the term error, as equity investment can move from company equity to debt. It is possible to include strong time-based individuality (unobserved specific effects, such as demographics and locations, through descriptive variables). Self-correlation occurs when a measured variable is delayed. The second reason for measuring a firm ($n = 19$) after a short-term measurement ($t = 6$) of a panel dataset is that simultaneous error control capacity allows some predicting variables to be endogenous. Due to the dynamic gathering of regression computation, the company's unique dummy variables cannot be used to control the firm's precise impact.

Level and differential equations have been combined in system GMM estimator. Level equations take advantage of the regressors' rear differential as an additional tool. One-step and two-step generalized methods of moment estimators were used here. Due to the practice of the maximum weight matrix, a two-step estimate is much more effective than a one-step estimation. A small cross-section measurement may (i) lead to influenced standard errors, (ii) affect estimate parameters, and (iii) lead to a poor exceptional identification test due to influenced standard errors [37]. Device growth, according to [38], is to blame for these problems. It is a solution that reduces instrumental combination measurement. It has been shown by [39,40] that if the dependent and explanatory variables are known and run continuously over time or behave nearly randomly, the variance of these components in differences performs poorly in regression [41]. This may be attributed to the parameter union's autoregressive approximation or an increase in the unpredictability of the individual effect owing to changes in idiosyncratic error. As a result, ref. [42] proposed a generalized method of moments technique that combined differences and regressions across levels to minimize the possible inaccuracy and obstacles associated with difference estimators.

To calculate the regression of differences, lagged differences (transformed) served as the means for regression in levels, and the generalized method of moments' estimate reliability was dependent on two diagnostics tests: one descriptive and one functional. It was legitimate to apply Sargent's tests for excessive instrument limitations because failing to discard the hypothesis meant that the instruments used in the model were accurate. The error term was put through a series of correlation tests [43]. Therefore, we had to rule out the existence of a 2nd order serial link even if there was no first-order autocorrelation (AR1) and reject $H_0$ (AR2). Using a two-step generalized method of moments estimate variable variance–covariance matrix (VCE) and a careful derivation of this restricted sample bias, ref. [44] defined the VCE (robust). Heteroscedasticity has little effect on reliable estimations of what has been rectified. When errors are heteroskedastic, the command estat sargan is not given following the description of the VCE, according to the Sargan test's output (robust). After establishing the VCE, an improved version of the Arellano–Bond test for autocorrelation was created (robust).

## 4. Results and Discussion

### 4.1. Descriptive Statistics

Table 1 provides a review of many variables that were considered. The following table shows the results of the variables of the study. The average standard deviation is 4.706 with a maximum price of 13.577. The average fixed asset revaluation surplus is 13.45, with a standard deviation of 2.363 and a maximum value of 16.02. According to the data, there is an average debt-to-asset ratio of 0.584 with a standard deviation of 0.322%. When it comes to investing for the long term, the average profit is 4.676 percent with an average standard deviation of 5.293 and a maximum of 14.477 percent. The standard deviation of the exchange rate is 0.039 or 4.676. The average cash-to-income ratio is 5.749 and the standard deviation is 10.573. In other words, the price ratio is 5.749 with a standard deviation of 57.821. The proportion of asset turnover is often expressed as an average of 0.801, with a standard deviation of 0.459.

### 4.2. Inferential Statistics

Table 2 supplies the variables with a Pearson correlation matrix. The dependent variable has a strong correlation with the operating income performance indicator. As a result, the model based on business performance accounting may validate crucial and direct connections between macroeconomic variables, such as the exchange rate (REER). When it comes to the relationship between the exchange rate and company performance, this finding agrees with that of [45]. The exchange rate, a macroeconomic variable based on outputs, has a strong connection to operating revenue. A variance inflation analysis (VIF) was utilized to provide good findings for a multicollinearity issue. A total of 1.87 was found, with the greatest VIF value being related to the variance impact factor in all models. The findings confirmed this. As a consequence, the specified variables provided no multicollinearity problem. Table 3 shows the results, which provide a rough idea of how much of an impact operative income has on company success. Table 3 shows the findings of the further investigations that were used to get strong evidence for the connection and to explain the hypothesis' results.

**Table 2.** Correlation analysis.

|  | OI | EXR | LTI | REFA | DAR | CIR | PER | ATR | ROA |
|---|---|---|---|---|---|---|---|---|---|
| OI | 1.000 | | | | | | | | |
| EXR | −0.223 *** | 1.000 | | | | | | | |
| LTI | 0.038 *** | 0.022 *** | 1.000 | | | | | | |
| RVFA | 0.067 *** | −0.087 *** | 0.178 *** | 1.000 | | | | | |
| DAR | 0.804 *** | −0.290 *** | −0.145 *** | 0.082 *** | 1.000 | | | | |
| CIR | 0.068 *** | −0.297 *** | 0.058 *** | 0.121 *** | 0.156 *** | 1.000 | | | |
| PER | 0.021 *** | −0.287 *** | 0.127 *** | 0.109 *** | 0.019 *** | 0.826 *** | 1.000 | | |
| ATR | −0.166 *** | 0.234 *** | −0.008 *** | −0.214 *** | −0.227 *** | −0.081 *** | −0.079 *** | 1.000 | |
| ROA | 0.133 *** | 0.018 *** | −0.126 *** | −0.061 *** | 0.102 *** | −0.423 *** | −0.261 *** | −0.206 *** | 1.000 |

*** $p < 0.01$, and OI = operating income (gross income − operative expenses); EXR = exchange rate, i.e., REER (real effective exchange rate); LTI = long-term investment; REFA = re-valuation of fixed assets; DAR = debt-to-assets ratio (short-term and long-term liabilities)/(noncurrent assets + current assets); CIR = cash-to-income ratio (operating cash flow/net profit); PER = price-earnings ratio (market price per share/basic earnings per share); ATR = assets turnover ratio (net sales/total assets); ROA = return on assets {net profit/(current assets plus noncurrent assets)}; EXR × DAR = REER (real effective exchange rate × debt-to-assets ratio).

**Table 3.** Linear regression model.

|  | (1) | (2) | (3) | (4) | (5) | (6) |
|---|---|---|---|---|---|---|
| **Variables** | OLS | RE | FE | PCSE AR1 | PCSE AR1 HetOnly | 2StepSys GMM |
| OI = L | | | | | | 0.048 * |
| | | | | | | (0.025) |
| EXR | 0.159 | 0.159 | 0.668 | 0.289 | 0.289 | 3.233 *** |
| | (7.265) | (7.265) | (5.739) | (6.043) | (6.137) | (1.157) |
| LTI | 0.127 *** | 0.127 *** | −0.149 | 0.125 *** | 0.125 *** | 0.128 ** |
| | (0.045) | (0.045) | (0.111) | (0.031) | (0.048) | (0.064) |
| REFA | −0.036 | −0.036 | −0.317 ** | −0.076 | −0.076 | −0.572 *** |
| | (0.113) | (0.113) | (0.130) | (0.193) | (0.182) | (0.017) |
| DAR | 12.34 *** | 12.34 *** | 15.25 *** | 13.18 *** | 13.18 *** | 0.578 |
| | (0.888) | (0.888) | (0.836) | (1.290) | (0.918) | (0.435) |
| CIR | −0.071 | −0.073 | −0.135 *** | −0.101 ** | −0.101 ** | −0.068 *** |
| | (0.049) | (0.049) | (0.042) | (0.040) | (0.047) | (0.014) |
| PER | 0.011 | 0.011 | 0.014* | 0.0119 * | 0.012 * | 0.005 *** |
| | (0.008) | (0.008) | (0.008) | (0.007) | (0.007) | (0.002) |
| ATR | 0.306 | 0.306 | 0.818 | 0.520 | 0.520 | 0.933 ** |
| | (0.610) | (0.610) | (0.909) | (0.699) | (0.701) | (0.472) |
| ROA | 0.036 | 0.036 | 0.007 | 0.030 | 0.031 | 0.212 *** |
| | (0.066) | (0.066) | (0.069) | (0.044) | (0.069) | (0.030) |
| Constant | 1.100 | 1.100 | 2.149 | 0.465 | 0.465 | 1.692 |
| | (33.95) | (33.95) | (26.84) | (28.08) | (28.73) | (5.355) |
| Observations | 113 | 113 | 113 | 113 | 113 | 95 |
| R-squared | 0.684 | 0.684 | 0.828 | 0.633 | 0.633 | |
| Number of firms | 19 | 19 | 19 | 19 | 19 | 19 |

| Diagnostic Checks | |
|---|---|
| Breusch and Pagan LM test for random effects | (8) *** |
| Hausman test | (18) *** |
| Multicollinearity test (VIF) | 1.87 |
| Heteroskedasticity test | 333.59 *** |
| Wooldridge test | 11.583/(0.0032) |
| Sargan test chi2(9)/($p$-Value) | (7.178) (0.618) |
| Arellano Bond Test AR (1) (Z) $p$-Value | (−1.8675) (0.0618) |
| AR (2) (z) $p$-Value | (1.557) (0.1195) |

Standard errors in parentheses *** $p < 0.01$, ** $p < 0.05$, * $p < 0.1$.

The revaluation of a company's fixed assets may be required to reflect the true value of its capital assets. The revaluation surplus was included in equity since capital assets such as equipment, plant, and property directly contribute to revenue generation and are transferred to retained earnings. REFA has an inverse effect on operative income while long-term investment has a positive relation. A revaluation of fixed assets, as described by the OLS and random effects models, has a negative impact on firm operative income according to regression results. With the help of the Hausman test, it is observed that the fixed effects model is more consistent than the random effects model. Using the system generalized method of moments, it is observed that the reevaluation of fixed assets has a negative and significant relationship at a one percent significance level with an increase in operating income. There is a significant association occurring between debt-to-assets ratios and operative income. The long-term investment in the fixed effects model has a negative significant relationship at a 5% significance level and in the system generalized method of moments at a 5% significance level. Therefore, this model is fit, and it can be further analyzed that for the OLS, the R-squared shows a 68.40% effect dependent on the variable, while the fixed effects model shows that all independent variables have an 82.80% significance in the dependent variable. Furthermore, the fixed effects model R-squared shows that all independent variables collectively significantly affect 82% of the dependent variables.

The linear equation explains the exchange rate coefficient with a significance level of 3.233 < 0.001. These findings suggest that the exchange rate influences the decision to reassess fixed assets. Thus, H1 supports the stance of this research work. Further, it is explaining the long-term investment coefficient with a significance level of 0.128 < 0.005. These findings suggest that the LTI influences the decision to reassess fixed assets and manage the business operations. Thus, H2 supports the stance of this research work. The fixed asset revaluation strength coefficient is $-0.572$ with a significance at a 1% level; this shows that the fixed asset intensity influences the decision to reevaluate fixed assets so H3 is supported. The coefficient of the debt-to-assets ratio is 0.578 with insignificance at more than 10% level [46]. The cash-to-income ratio coefficient is $-0.068$, meaning a cash holding decision negatively affects operational income [47], thus concluding that H5 is supported. The price-earnings ratio coefficient is 0.005 at a 5% significant level. This means it has a significant positive influence on operating income [48]. The assets turnover ratio coefficient also has a positive sign, meaning it is significantly enhancing the operating income of the selected sugar firms [49]. Therefore, H7 is supported. ROI and operational income both are performance measures that are interlinked, so they have positive coefficients and support each other [50]. Using the panel-corrected standard errors (PCSE) of the model to gain robustness and using the system generalized method of moments model helped us to manage the heteroskedasticity problem which existed in the FE model exposed through the heteroskedasticity test by using the httest3 command results in Stata's output. The Woolridge test also depicted that a serial correlation problem existed in the model. This serial correlation and heteroskedasticity problems were resolved by the panel-corrected standard errors in a static panel, and the dynamic panel was covered by the system generalized method of moments which was the basic purpose of using the system generalized method of moments model.

The long-term investment of the company affects the operating profits. This finding shows that even if a business is likely to take several actions to handle the long-term investment more accurately, modest profits would minimize the devaluation of fixed assets and affect its future expenditure because of the introduction of operating expenses (deprecation costs) from the revaluation of assets, thereby reducing the attention of investors, media, administration, and authorities. This finding indicates, besides differences in the market and cultural climate, that the incentive of management in Pakistan to revalue assets remains consistent with the behavior taken by managers in Pakistan. This result means that investors consider revaluation to be value-relevant, which can be used for investment

choices because assets revaluation behavior can minimize knowledge asymmetry amongst investors and the management of companies.

In Table 4, we chose for inferences model 6, where the two-step system GMM outputs explored the nonlinear model depicting the interaction variable as having a significant value, meaning it had a moderating effect between the dependent variable and independent variable, such as operative income and exchange rate, and debts-to-assets ratio [51,52]. Further, it is special that the lag-dependent variable also has a significant value, which means the dynamic model has its significance and is properly applied in this context [37,53,54]. The REFA coefficients value is negative, which means it has negative impact on firm's operative incomes, while long-term investment has a positive impact on firms' operative incomes. Basically, we applied OLS, RE, and FE to make the inference but the results were not consistent compared to those of PCSE. Ultimately, we decided to apply the two-step system GMM to cover the problem of heteroscedasticity, serial correlation, and normality issue of the data. Additionally, the discrepancy is more obvious when the data set's temporal dimension T is small. N would not benefit from a bigger cross-section dimension since the number of unobserved effects directly affects N's growth. Furthermore, the system GMM may be utilized as a benchmark if the following two conditions are met: the initial lag of the dependent variable is approximated by the system GMM between the OLS and fixed effects measurements, and the instruments used for the differenced computation and level equation are accurate. According to [40], the system GMM estimator is more effective at minimizing the finite-sample biases of the difference GMM estimator. Difference and system GMM have the drawback of being intricate estimators of variable performance. As a result, there is a higher chance that these estimators will be used incorrectly. In this investigation, the system GMM estimator was used. The logic of the instruments has a vital influence on the correctness of the GMM system decision. The validity of the instrument was thus evaluated using the Sargan test. The residuals were examined for the presence of second-order serial interaction using the Arellano–Bond AR (2) test.

**Robustness test and endogeneity problem**

Thus, the problem of endogeneity emerges when we have a variable Z that is connected to Y, but also related to X and excluded from the model. There are several causes of endogeneity, such as if the seller changes location depending on time or pricing depending on the length of the line. The origins of endogeneity may be categorized generically as: Example 1—omitted variables (an illustration is ice cream, in which the temperature is the omitted variable, and the price is the endogenous variable). Example 2—where X leads to Y and Y leads to X. This is both the most typical and the most challenging. If the structural information is known, it is simple to deal with missing variables. In Example 1, if the vendor modifies the price depending on the length of the queue as a control (as explored in dynamic programming or sequential control theory), there is simultaneity as the queue causes the price and the price causes the queue. Self-selection may also lead to simultaneity. Example 3—preference bias: if sampling is skewed, we cannot assess the impact of therapy on a person.

Prior work found that the association concerning a revaluation of fixed assets and organization performance (operative earnings) was an ever-present problem. To control for potential endogeneity, pooled least squares with panel-corrected standard errors methods were used. Following prior investigations, the most basic conducive variables were used as the first lag of operating income signs, such as L.OI [55]. As a result, the OI's first leg was used as a model requirement. According to the results of the linear model, all outcomes were the same as in the nonlinear model. When additional variables help establish a link between an intervention and an outcome, the overall correlation cannot be interpreted as having a causal effect.

When the explanans (X) may be impacted by the explanandum (Y), or when both may be jointly influenced by an unmeasured third, the fundamental issue of endogeneity arises. One component of the more general issue of selection bias is the endogeneity problem. The endogeneity issue is a term that has been used to describe this situation. The idea behind

the phrase is that a treatment cannot be thought of as being independently determined from the result-determining model, as it should be, but rather as being jointly determined with outcomes by a third factor endogenously occurring inside the outcome-dependent model. Due to the problem of endogeneity, simple correlation-based assessments of the relationship between treatment and result will be erroneous estimates of the influence of treatment causation on outcomes. There seems to be no endogenous issue in selecting the operating income model based on accounting as a result, and the basic premise has been completely validated. EXR (REER) has a substantial impact on operating income performance, even when endogeneity is taken into account in the model and tested for. The endogeneity issue was covered by panel-corrected standard errors methods [56] and the system generalized method of moments method, which detects a potentially eliminated variable error [57].

**Table 4.** Nonlinear regression model.

| Variables | (1) OLS | (2) RE | (3) FE | (4) PCSE AR1 | (5) PCSE Hetonly | (6) Twostep Sys GMM |
|---|---|---|---|---|---|---|
| OI = L | | | | | | 0.0810 *** |
| | | | | | | (0.0158) |
| EXR | −28.07 * | −28.07 * | −40.99 *** | −32.13 | −32.13 * | 26.01 *** |
| | (16.96) | (16.96) | (14.19) | (19.94) | (18.40) | (3.121) |
| LTI | 0.123 *** | 0.123 *** | −0.142 | 0.122 *** | 0.122 *** | 0.170 *** |
| | (0.0446) | (0.0446) | (0.105) | (0.0251) | (0.0460) | (0.0612) |
| REFA | −0.0349 | −0.0349 | −0.392 *** | −0.0762 | −0.0762 | −0.577 *** |
| | (0.112) | (0.112) | (0.126) | (0.207) | (0.189) | (0.0163) |
| DAR | −189.9 * | −189.9 * | −275.0 *** | −215.3 * | −215.3 ** | 157.0 *** |
| | (110.0) | (110.0) | (91.24) | (114.5) | (109.4) | (18.97) |
| CIR | −0.0745 | −0.0745 | −0.143 *** | −0.105 *** | −0.105 ** | −0.0477 *** |
| | (0.0488) | (0.0488) | (0.0406) | (0.0408) | (0.0453) | (0.0119) |
| PER | 0.00990 | 0.00990 | 0.0126 * | 0.0113 | 0.0113 * | 0.00477 *** |
| | (0.00825) | (0.00825) | (0.00694) | (0.00703) | (0.00645) | (0.00158) |
| ATR | 0.364 | 0.364 | 0.904 | 0.566 | 0.566 | 1.680 *** |
| | (0.604) | (0.604) | (0.865) | (0.650) | (0.704) | (0.273) |
| ROA | 0.0388 | 0.0388 | −0.0292 | 0.0267 | 0.0267 | 0.223 *** |
| | (0.0649) | (0.0649) | (0.0667) | (0.0473) | (0.0700) | (0.0270) |
| EXDAR | 43.31 * | 43.31 * | 62.12 *** | 48.90 ** | 48.90 ** | −33.36 *** |
| | (23.57) | (23.57) | (19.52) | (24.48) | (23.37) | (4.076) |
| Constant | 133.0 * | 133.0 * | 198.1 *** | 152.1 | 152.1 * | −106.5 *** |
| | (79.23) | (79.23) | (66.66) | (93.20) | (86.19) | (14.56) |
| Observations | 113 | 113 | 113 | 113 | 113 | 95 |
| R-squared | 0.6593 | 0.6937 | 0.846 | 0.652 | 0.652 | |

| Diagnostic Checks | |
|---|---|
| Breusch and Pagan LM test for random effects | (9) *** |
| Hausman test | (18) *** |
| Multicollinearity test (VIF) | 1.86 |
| Heteroskedasticity test | 255.26 *** |
| Wooldridge test | 10.423/(0.0047) |
| Sargan test chi2(9)/(p-Value) | (11.77) (0.462) |
| Arellano Bond Test AR (1) (Z) *p*-Value | (−1.658) (0.095) |
| AR (2) (z) *p*-Value | (1.278) (0.201) |

Standard errors in parentheses *** $p < 0.01$, ** $p < 0.05$, * $p < 0.1$.

## 5. Conclusions

This research article was written to examine the impact of revaluation and its analysis with fixed assets on prospective stable performance in the sugar sector in Pakistan. Our findings indicated that the evaluation had a major impact on the growth of firms, which means that the recovery strategy of a company does not reverse the overall effect of the

decline in asset prices, which is actually due to a decrease. The primary reason for this is that investors do not see the value of assets increase as much as they are strongly associated with compact firms' performance. The investigation of this same increase in income leads to an increase in operating revenue. Investors are sufficiently intelligent and have a direct exposure to ensure precise information that enables them to understand the asset value reality. Thus, although the profit is repaid, the operating income, which has a positive impact on the market, has a positive effect on the equity ratio. In line with these outcomes, this study will help companies understand that it is not possible to evaluate assets at reasonable prices and spend more on reviews or audits. The main limitations of this study were the time constraints. To generalize our results, this study should be extended to other areas. In addition, increasing the interval will have a better effect and will remove the main obstacle of the study, so that better results can be obtained.

*Contribution/Origin*

This research was one of the very rare investigations were the results of macroeconomic variable EXR (REER), as a moderator between operating income and the firm's DAR of various sugar firms in Pakistan, were examined. Prior investigations were focused on specific areas, such as the food and textile business, and investigated the impact of macroeconomic determinants on firm performance or firm value. We provided information on this issue in various sectors, including all manufacturing industries. Here, the system generalized method of moments and the new analysis technique of panel-corrected standard errors was applied to make inferences.

**Author Contributions:** Conceptualization: S.H. (Sarfaz Hussain), M.E.H. and P.S. Methodology: S.H. (Sarfaz Hussain), P.S., W.A.W. and P.M. Software: S.H. (Samina Haque), S.H. (Sarfaz Hussain), M.E.H. and P.S. Validation: S.H. (Sarfaz Hussain), M.E.H. and P.S. Formal analysis: S.H. (Sarfaz Hussain), P.S., W.A.W. and P.M. Investi-gation: S.H. (Sarfaz Hussain), M.E.H. and P.S. Resources: S.H. (Sarfaz Hussain), M.E.H. and P.S. Data cu-ration: S.H. (Samina Haque), S.H. (Sarfaz Hussain), M.E.H. and P.S. Writing—original draft prepara-tion: S.H. (Sarfaz Hussain), M.E.H. and P.S. Funding acquisition: P.S. Project administration: S.H. (Sarfaz Hussain) and W.A.W. Supervision: mutual teamwork. All authors have read and agreed to the published version of the manuscript.

**Funding:** This manuscript received no external funding.

**Institutional Review Board Statement:** Not applicable.

**Informed Consent Statement:** Not applicable.

**Data Availability Statement:** Data available on request from the first author.

**Acknowledgments:** This research was supported by Perengki Susanto, Universitas Negeri Padang, Padang, Indonesia, Department of Management, Jalan Hamka, Freshwater Padang, West Sumatra, Indonesia. We are thankful to Perengki Susanto. We thank our colleagues team members those provided insight and expertise that greatly assisted the research. We would also like to show our gratitude to the (Atiya Yusuf Statistical Officer, Statistics & Data ware House Department, State Bank of Pakistan. Phone: +9221-33114997, Email: atiya.yusuf@sbp.org.pk) for sharing their pearls of wisdom with us during the course of this research, and we thank 2 reviewers for their insights.

**Conflicts of Interest:** Authors declare no conflict of interest.

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
