# Peer review of "The Quality of Fair Revaluation of Fixed Assets and Additional Calculations Aimed at Facilitating Prospective Investors’ Decisions"

_sustainability, doi:10.3390/su141610334_

Round 1

Reviewer 1 Report

This is very interesting article that reveals the issues of investments and the analysis of financial statements. However, the Abstract does not fully disclose the relevance and purpose of the study. The keywords do not correspond to the meaning of the study, for example, it is not clear where "Jel Classifications: M480, M410, M160" is used?

I would like to note that the article introduction and the second part provide a very good literature review, however, the chosen research method is not substantiated, and some sources are already outdated (more than 10 years), there are errors in the reference list formatting.

The third section presents the research's methodology, however, its structure is not presented. In paragraph 3.2. the measurement values, their abbreviation ​​are described. Authors said that measurement values are presented in table 1, however, these measurement values ​​are used in all other tables 2-4 and after these tables, the descriptions of these measurement values ​​are repeated each time. It may be worthwhile in paragraph 3.2 to immediately say that all measurement values ​​​​are presented in all tables and remove the repeated transcripts after each table. In paragraph 3.3, where the model is presented, the choice of the variable “Operating income (OI)” is not justified, formula (1) is not clear and it is not clear how all the presented formulas are used in further research, there is no application logic.

There is no section in the study where the study results are presented. It is necessary to redo the structure of paragraph 3, where it is necessary to leave the research method, variables and objects of measurement and models, and the research results themselves in the form of tables, diagnostic checks should be separated into the 4th section "Results".

The last section presents the main research findings.

The article contains significant errors:

1. There are a lot of abbreviations in the article that need to be deciphered: FASB, IASB, IAS, ROA, SBP, GMM, PCSC, PCSE.

2. Line 39: the authors of literature sources are incorrectly indicated.

3. Line 172: "AIT = Assets Turnover Ratio" is incorrectly indicated, "ATR" is indicated in all formulas and tables.

4. Line 167: “REFA = Re-Evaluation of Fixed Assets” is indicated, but in the tables “RVFA”.

5. Line 184: invalid reference to formulas.

6. Line 187: the explanation of the index must be indicated in small Latin letters, according to the presented formulas 3 and 4, mathematically correctly this explanation can be written as follows   .

7. Line 207-208: it is necessary to correctly format these explanations “XIT is 1*K, and β is K*1. µit. µit = λi + Ɛit".

8. Line 219: Need clarification "XIT is equal to 1*and 1*".

9. Line 236: the authors of the literature source (ii), (iii) are not indicated.

10. Line 253: The authors of Arellano and Bond, 1991 are not listed.

11. In the list of references 1 and 2 the source is the same.

The article can be recommended for publication after the above remarks are eliminated. The article carried out a large amount of work, which deserves special scientific interest.

Author Response

Reviewer 1 | 28 Jul 2022 | Manuscript ID: sustainability-1822948

This is very interesting article that reveals the issues of investments and the analysis of financial statements.

  1. There are a lot of abbreviations in the article that need to be deciphered: FASB, IASB, IAS, ROA, SBP, GMM, PCSC, PCSE.

Author’s reply: Thank you so much for suitable corrections. We have carefully done the required changes. It is now in full text form instead of abbreviations.  

  1. Line 39: the authors of literature sources are incorrectly indicated.

Author’s reply: it is corrected

  1. Line 172: "AIT = Assets Turnover Ratio" is incorrectly indicated, "ATR" is indicated in all formulas and tables.

Author’s reply: it is corrected

  1. Line 167: “REFA = Re-Evaluation of Fixed Assets” is indicated, but in the tables “RVFA”

Author’s reply: it is corrected

  1. Line 184: invalid reference to formulas.

Author’s reply: it is corrected

  1. Line 187: the explanation of the index must be indicated in small Latin letters, according to the presented formulas 3 and 4, mathematically correctly this explanation can be written as follows ? = 1, …, 19 or ? = 1̅̅,̅19̅̅̅.

Author’s reply: Corrected according to suggestions

  1. Line 207-208: it is necessary to correctly format these explanations “XIT is 1*K, and β is K*1. µit. µit = λi + Ɛit".

Author’s reply: format Corrected according to suggestions

  1. Line 219: Need clarification "XIT is equal to 1*and 1*"

Author’s reply: clearly address

 Line 236: the authors of the literature source (ii), (iii) are not indicated.

  1. Author’s reply: removed and makes updated literature
  2. Line 253: The authors of Arellano and Bond, 1991 are not listed.

Author’s reply: right now, listed

  1. In the list of references 1 and 2 the source is the same.

Remove one of them

The article can be recommended for publication after the above remarks are eliminated. The article carried out a large amount of work, which deserves special scientific interest

Reviewer 2 Report

In light of the upcoming economic crisis, the topic of the article is topical and interesting. It can give a lot of guidance to smaller and larger investors in making decisions about buying new assets. The research has been narrowed down to one industry and one country, but the literature considerations also include an assessment of other countries in terms of assets held and the investment decision. Moreover, the choice of Pakistan as a research entity has been properly justified. The research period until 2018 requires the preparation of recommendations or forecasts for the current years for the industry under study in the period of the economic crisis, which did not occur in the analyzed period. I believe that the Data sections can be combined with the Variables section. Data sources are missing from this section. How was the information for the study collected? Please correct the font next to the patterns, also edit the editor. Research methodology should also be added in the article. "Robustness test and endogeneity problem" is probably a discussion of the test results? I'm not sure, but other studies have been given in this regard. If it's a discussion, it needs to be developed.

Author Response

Reviewer 2 | 28 Jul 2022 | Manuscript ID: sustainability-1822948

  1. Are the research design, questions, hypotheses and methods clearly stated?

Author’s reply: yes, it is stated clearly, and get improved according to instructions  

  1. For empirical research, are the results clearly presented?

Author’s reply: improved according to instruction

  1. Are the conclusions thoroughly supported by the results presented in the article or referenced in secondary literature?

Author’s reply: yes it is verified and get improved

Round 2

Reviewer 1 Report

Accept in present form

Author Response

I see everywhere in the comments you selected the yes option I think after the first review nothing to revise.

I have already addressed these issue which was highlighted 

Reviewer 1 | 12 August 2022 | Manuscript ID: sustainability-1822948

This is very interesting article that reveals the issues of investments and the analysis of financial statements.

  1. There are a lot of abbreviations in the article that need to be deciphered: FASB, IASB, IAS, ROA, SBP, GMM, PCSC, PCSE.

Author’s reply: Thank you so much for suitable corrections. We have carefully done the required changes. It is now in full text form instead of abbreviations.  

  1. Line 39: the authors of literature sources are incorrectly indicated.

Author’s reply: it is corrected

  1. Line 172: "AIT = Assets Turnover Ratio" is incorrectly indicated, "ATR" is indicated in all formulas and tables.

Author’s reply: it is corrected

  1. Line 167: “REFA = Re-Evaluation of Fixed Assets” is indicated, but in the tables “RVFA”

Author’s reply: it is corrected

  1. Line 184: invalid reference to formulas.

Author’s reply: it is corrected

  1. Line 187: the explanation of the index must be indicated in small Latin letters, according to the presented formulas 3 and 4, mathematically correctly this explanation can be written as follows ? = 1, …, 19 or ? = 1̅̅,̅19̅̅̅.

Author’s reply: Corrected according to suggestions

  1. Line 207-208: it is necessary to correctly format these explanations “XIT is 1*K, and β is K*1. µit. µit = λi + Ɛit".

Author’s reply: format Corrected according to suggestions

  1. Line 219: Need clarification "XIT is equal to 1*and 1*"

Author’s reply: clearly address

 Line 236: the authors of the literature source (ii), (iii) are not indicated.

  1. Author’s reply: removed and makes updated literature
  2. Line 253: The authors of Arellano and Bond, 1991 are not listed.

Author’s reply: right now, listed

  1. In the list of references 1 and 2 the source is the same.

Remove one of them

The article can be recommended for publication after the above remarks are eliminated. The article carried out a large amount of work, which deserves special scientific interest

Reviewer 2 | 28 Jul 2022 | Manuscript ID: sustainability-1822948

  1. Are the research design, questions, hypotheses and methods clearly stated?

Author’s reply: yes, it is stated clearly, and get improved according to instructions 

  1. For empirical research, are the results clearly presented?

Author’s reply: improved according to instruction

  1. Are the conclusions thoroughly supported by the results presented in the article or referenced in secondary literature?

Author’s reply: yes it is verified and get improved
